# Circulating and Tumor-Associated Neutrophils in the Era of Immune Checkpoint Inhibitors: Dynamics, Phenotypes, Metabolism, and Functions

**DOI:** 10.3390/cancers15133327

**Published:** 2023-06-24

**Authors:** Lara Gibellini, Rebecca Borella, Elena Santacroce, Eugenia Serattini, Federica Boraldi, Daniela Quaglino, Beatrice Aramini, Sara De Biasi, Andrea Cossarizza

**Affiliations:** 1Department of Medical and Surgical Sciences for Children and Adults, University of Modena and Reggio Emilia, 41121 Modena, Italy; rebeccaborella7993@gmail.com (R.B.);; 2Department of Life Sciences, University of Modena and Reggio Emilia, 41125 Modena, Italy; 3Division of Thoracic Surgery, Department of Medical and Surgical Sciences (DIMEC), University Hospital GB Morgagni—L Pierantoni, 47121 Forlì, Italy

**Keywords:** neutrophils, tumor microenvironment, metabolism

## Abstract

**Simple Summary:**

Neutrophils are the most abundant leukocytes in the circulation, represent the first line of defense in the immune system and mediate inflammation. Increasing evidence suggests that neutrophils constitute a large population of cells with phenotypic and functional heterogeneity. In this review, we summarize and discuss new findings delineating that both circulating neutrophils and tumor-associated neutrophils have a role in tumor prognosis and resistance to immune checkpoint inhibitors.

**Abstract:**

Neutrophils are the most abundant myeloid cells in the blood and are a considerable immunological component of the tumor microenvironment. However, their functional importance has often been ignored, as they have always been considered a mono-dimensional population of terminally differentiated, short-living cells. During the last decade, the use of cutting-edge, single-cell technologies has revolutionized the classical view of these cells, unmasking their phenotypic and functional heterogeneity. In this review, we summarize the emerging concepts in the field of neutrophils in cancer, by reviewing the recent literature on the heterogeneity of both circulating neutrophils and tumor-associated neutrophils, as well as their possible significance in tumor prognosis and resistance to immune checkpoint inhibitors.

## 1. Introduction

The clinical value of immune checkpoint inhibitors (ICIs) has been reported across many cancer types, including metastatic melanoma, lung cancer, colorectal cancer, and urothelial carcinoma. Currently available ICIs, administered as standalone therapies typically lead to 20–40% objective response rates (ORRs) across cohorts of patients affected by different neoplasms. However, as most of these patients do not benefit from these therapeutic interventions, understanding the mechanisms and the biological processes underlying the responses and resistance to ICI therapy is therefore crucial to improve the clinical efficacy and durability of responses.

One the possible mechanisms relies on the capacity of immune cells, including neutrophils, which express the ligands of these checkpoint inhibitors and infiltrate the tumor microenvironment (TME), typically sustaining tumor progression [1]. Neutrophils have often been described as a short-living, homogenous, and mono-dimensional population, with restricted roles in tumor progression and responses to therapy. However, recent multi-omic and single-cell analyses have revealed that neutrophils are heterogeneous in terms of gene expression, transcription factors, genetic signature, and developmental paths [2,3,4]. Neutrophil heterogeneity has been reported in physiological conditions as well as in infections and cancer, being influenced, for instance, by gender, pregnancy, age, and the gut microbiome [2,5,6,7,8]. Neutrophil heterogeneity is present in tissues and in the peripheral blood where three main transcriptionally distinct neutrophil subsets can be observed [7]. Among these, a subset of mature neutrophils expresses a set of interferon-stimulated genes (ISGs), including ISG15, an interferon-induced protein with the tetratricopeptide repeats (IFIT) 1 and 2, suggesting that these cells could be primed to counteract pathogens even before infection takes place [7]. The presence of the different clusters of circulating neutrophils has been reported for several pathological conditions and clinical settings, including infections, sepsis, and cancer [9,10]; in addition, the fact that they migrate to tissues and can also be massively recruited into the sites of sterile damage has recently challenged the notion that these cells are exclusively antimicrobial, thereby raising the possibility that they actively take part in other subtle processes. For example, concerning cancer, tumor-associated neutrophils (TANs) exhibit both pro-tumorigenic and anti-tumorigenic properties [2,11]. However, there is little information on how the TANs heterogeneity is established and maintained, and, most importantly, pivotal studies rely on murine models, but are scarce on humans.

## 2. Neutrophils

### 2.1. Features and Functions

Neutrophils represent 50–70% of circulating leukocytes in humans and are the body’s first essential line of defense against infections. They are the main effectors of acute inflammation, but they can also contribute to chronic inflammatory conditions and to adaptive immune responses [12,13]. Granulocyte-colony stimulating factor (G-CSF) is a crucial regulator of neutrophil release from the bone marrow and of neutrophil biology [14,15]. The lack of G-CSF receptors causes neutropenia in both mice [16,17] and humans [18,19]. Approximately 10^11^ neutrophils are generated per day in normal human adults, at the steady-state, but they increase dramatically by the actions of the G-CSF during infection [14,20].

Neutrophil differentiation is a complex and heterogeneous process and arises from the bone marrow (BM). It starts from hematopoietic stem cells (HSCs), which, following their differentiation into common myeloid progenitors (CMPs), give rise to the granulocyte-monocytes progenitors (GMPs) [15,21]. The subsequent stages include promyelocytes, myelocytes, metamyelocytes, and finally the banded and segmented nucleus neutrophils. GMPs differentiate into promyelocytes, which express the neutrophil lineage marker CD66b [15,21]. Subsequent neutrophil development occurs through the upregulation of CD11b and CD16. Promyelocytes differentiate into myelocytes and metamyelocytes, and lastly into banded (immature) and segmented (mature) neutrophils [14]. The traditional classification of neutrophil differentiation based on appearance may not represent the full process. Over the past decade, the use of single-cell technologies allowed for a better definition of the neutrophil maturation stages. In mice, GMPs develop into the pro-neutrophils proNeu1, which develop into the intermediate progenitors proNeu2, which in turn mature into the highly proliferative precursors termed pre-Neu. The last ones share a transcriptional profile with the unipotent neutrophil progenitors (NeP), which, as described by Zhu et al., have little mobility and are most plentiful in the BM. They undergo further differentiation into non-proliferating immature neutrophils and subsequently mature neutrophils, which are predominantly present in the blood, have lost their proliferative capacity, gained a high motility, and mediate effector functions [11,22]. Several studies have proposed that GMPs consist of several heterogeneous myeloid progenitor cells, including neutrophil progenitors, rather than a single homogeneous cell type. However, these studies do not exclude the possibility that an ‘earlier’, still undefined, progenitor exists that could solely give rise to neutrophils and monocytes [7,23,24,25,26,27,28]. The high degree of neutrophil heterogeneity has been observed in cancer and is associated with disease progression. A typical hallmark of cancer is the egress of neutrophils from the bone marrow at the earlier stages. It was observed that immature neutrophils with a banded nuclear morphology play a pro-tumorigenic role [11]. In lung cancer patients, developmental stages even earlier than NeP/preNeu have been observed in both the blood and tumor [24]. In melanoma patients, circulating NePs are increased compared to healthy subjects, and are able to sustain tumor growth and immunosuppression [28].

Neutrophil trafficking from the bone marrow to the peripheral blood is tuned by signaling through the CXC-chemokine receptors (CXCR)-2 and CXCR4, both in the human and mouse [29,30,31]. In particular, the expression of CXCR4 retains immature neutrophils within the bone marrow, while its decreased expression in mature neutrophils, together with the activation of CXCR2 signaling, triggers the entry of these cells into the blood [32]. Subsequently, once neutrophil functions are accomplished, the process of aging begins under the control of various transcriptional programs, which are also cell intrinsic. The production of CXCL2 by neutrophils and its binding to CXCR2 induces neutrophil aging and the upregulation of CXCR4, which, in turn, drives neutrophil homing back into the bone marrow and clearance by macrophages [2,31,33]. In mice, the extrinsic and intrinsic mechanisms involved in aging have been proposed to be temporally regulated with a neutrophil-intrinsic timer [29]. Diurnal changes in several genes related to the pathways of inflammation, migration, and apoptosis have been identified at multiple times in the circulating neutrophils. The circadian program of aging enables the early activation of neutrophils which helps in anticipating infections [29]. However, an excessive activation when the risk of infection is low can lead to vascular and tissue damages. The high prevalence of infections and cardiovascular diseases prompts us to understand whether the identification of diurnal programs in neutrophils could offer therapeutic alternatives for these life-threatening complications. In principle, targeting CXCR2 or CXCR4 with specific agonists might allow for the pharmacological and transient manipulation of the biological timer through either promoting defense or protecting the vasculature. Thus, for humans suffering from cardiovascular diseases, it might be desirable to block aging, whereas patients undergoing immunosuppressive therapies might benefit from drugs that promote aging, thus preventing exacerbated inflammation [29].

Mature neutrophils are ~7–10 µm in diameter, and are characterized with the typical lobulated nuclei, do not divide, and contain multiple granules and secretory vesicles in their cytoplasm (Figure 1) [14,15]. Neutrophil effector functions include phagocytosis, degranulation, and neutrophil extracellular trap (NET) formation, characterized with oxidative and non-oxidative processes [34,35]. Neutrophils express high level of nicotinamide adenine dinucleotide phosphate (NADPH) oxidase and myeloperoxidase (MPO). NADPH oxidase is a multi-subunit enzyme which catalyzes the formation of superoxide anion, that, in turn, mediates the neutrophils antimicrobial activity [35]. MPO catalyzes the conversion of hydrogen peroxide, obtained from superoxide ions and chloride ion to form hypochlorite, a harmful chemical [36,37,38]. Neutrophils also express enzymes involved in the synthesis of bioactive lipids [39], including leukotrienes and prostaglandins, which increase neutrophil chemoattraction, cytokine production, and phagocytosis during inflammation [39]. However, other lipids are capable to resolve neutrophil inflammation: the specialized pro-resolution mediators, such as resolvins, maresins, protectins, and lipoxins, which disrupt neutrophil chemotaxis and phagocytosis, and, consequently, efferocytosis, i.e., the clearance of apoptotic cells, which is essential for the maintenance of tissue homeostasis [39,40].

### 2.2. Metabolism

The influence of metabolism on neutrophil functions has only recently received appropriate recognition [41]. Glycolysis has long been viewed as the unique source of energy for these cells. Indeed, an impairment in glucose shuttling between the endoplasmic reticulum (ER) and the cytosol can lead to neutropenia and the deterioration of respiratory burst activity, ATP production, and bacterial killing, both in the human and mouse [42,43,44]. Neutrophil dependence upon glycolysis allows these cells to function and survive in inflamed sites, including in the TME, where limited oxygen availability may render oxidative metabolism to be ineffective in meeting the energy demand. Elevated levels of GLUT1 mRNA and protein have been reported in lung adenocarcinoma [45]. In particular, TANs exhibited a higher Glut1 expression with a subsequently enhanced glucose uptake, whereas the neutrophil-specific deletion of solute carrier family 2 member 1 (Slc2a1), which encodes for Glut1, was found to be associated with a decreased TAN survival and limited tumor growth in mice [45]. Hyperactivated glycolytic activity was also observed in a subset of neutrophils infiltrating pancreatic ductal adenocarcinoma, whose signature is an unfavorable prognostic factor, thus corroborating the importance of a glycolytic switch in the pro-tumor functions in neutrophils [46].

While glycolysis still remains the main metabolic pathway engaged in the neutrophil cytosol, recent studies have indicated that other routes also operate during their differentiation and function [47]. Neutrophils utilize the pentose phosphate pathway (PPP) to produce NADPH and ribose [48]. Recent findings have suggested that during acute oxidative stress, neutrophil metabolism is immediately reconfigured around the PPP, which is coupled to the oxidative burst. In this setting, human neutrophils switch from a glycolysis-dominant metabolism to a mode termed the ‘pentose cycle’, or oxidative (ox)-PPP, where glucose-6-phosphate is redirected into oxidative PPP, thus maximizing the NADPH yield to fuel reactive oxygen species (ROS) production through NADPH oxidase [49]. The inhibition of ox-PPP strongly prevents oxidative bursts, NET release, and pathogen killing [49]. With limited glucose availability, fatty acid oxidation can also be used to support NADPH oxidase-dependent ROS production [50]. In the presence of physiological stresses, including glucose depletion and hypoxia, or during pro-inflammatory activation, neutrophils can generate intra-cellular glucose stores in the form of glycogen, thus contributing to neutrophil functions and survival [51,52].

The accumulating evidence has suggested that fatty acid oxidation (FAO) is severely increased in blood and tumor polymorphonuclear myeloid-derived suppressor cells (PMN-MDSCs), and in granulocytic MDSCs. MDSCs include several myeloid cells at different stages of differentiation, with a strong immunosuppressive role in the TME. MDSCs exert a potent activity against T cells by depriving them from essential amino acids and reducing their expression of L-selectin. MDSCs also promote oxidative stress and the induction of immunosuppressive cells like regulatory T (Tregs) and T helper 17 (Th17) cells. In mice, MDSCs accumulate in several types of cancer, where they sustain cancer progression by inducing angiogenesis, tumor growth, and metastasis [53]. In contrast, preventing FAO can modulate the immunosuppressive functions of these cells [54].

A growing body of evidence has indicated that the function of mitochondria in neutrophils broadens beyond regulating apoptosis, and includes (but is not limited to) the control of neutrophil development, chemotaxis, ROS production, degranulation, and NET release [55]. Mitochondrial DNA is also part of the NETs, thus representing a powerful tool used by neutrophils to trap bacteria and contributing to the elimination of infection. In addition, neutrophil mitochondria are a crucial source of damage-associated molecular patterns (DAMPs), which promote inflammation when released into the extracellular milieu [55].

Thus, whether or not microenvironmental conditions are permissive, glycolysis remains one of the major metabolic pathways engaged by the neutrophils to assist their survival and functions. Nonetheless, when needed, neutrophils can adapt and shift to other metabolic pathways. However, how these variations occur, especially at the gene expression level, remains unclear.

### 2.3. Low-Density Neutrophils

Low-density neutrophils (LDNs) were first identified as a subset of neutrophils that layer in the same fraction of peripheral blood mononuclear cells (PBMCs) when leukocytes are separated through Ficoll–Hypaque density gradient centrifugation [56,57,58]. LDNs can be separated from mature high-density neutrophils (HDNs) isolated from the high-density fraction. Neutrophils in the low-density fraction possess a decreased phagocytic activity, impaired reactive oxygen species (ROS) production, and a diminished capacity to inhibit CD8+ T cell proliferation when compared to HDNs [59]. LDNs are scarcely present in healthy individuals but are elevated in various pathological conditions, including infections, cancer, and autoimmune diseases [60,61]. LDNs consist of mature (multilobed nuclei) and immature (band-shaped nuclei) neutrophil subsets, and have been likely associated with immunosuppressive functions, although the exact mechanisms at the basis of their functions are still not clearly elucidated (Figure 1) [62].

Since the first descriptions of LDNs, several attempts have been made to identify the specific cellular markers that are able to discriminate between the LDNs and normal-density neutrophils (NDNs). However, the markers used to identify LDNs have diverged between studies and between diseases, and a conclusive set has yet to be established. A recent report revealed that CD66b, CD16, CD15, CD10, CD54, CD62L, CXCR2, CD47, and CD11b are expressed at equal levels in the LDNs and NDNs from healthy individuals [63]. Other works described LDNs as CD25+CCR6+CD24+CD66b+CD11b+ in asymptomatic pregnant women infected with SARS-CoV-2 [60], and as CD66b+CXCR1+CCR6+ in a cohort of aged individuals with severe COVID-19 [61].

LDNs appear transiently in the blood, in self-resolving inflammation, and accumulate in tumor-bearing mice, as well as in patients with advanced cancer [62,64,65,66]. The accumulation of LDNs has been reported in the peripheral blood of patients with renal carcinoma, head and neck cancer, non-small cell lung cancer (NSCLC), pancreatic cancer, colon cancer, and breast adenocarcinoma [67,68,69,70]. Their frequency often appears to be correlated with disease aggressiveness and the treatment response. In fact, in cancer, LDNs are not capable in killing tumor cells and display immunosuppressive properties that support tumor progression [62]. LDNs can also promote tumor metastasis by releasing NETs within the tumor microenvironment (TME), which can entrap circulating tumor cells and drive them towards secondary niches [59].

Although LDNs have been extensively studied in various diseases, several aspects of their functions, characteristics, and phenotypes are still a matter of intense controversy. They further enrich the concept of neutrophilic heterogeneity, as pro-inflammatory LDNs have been described in autoimmunity, infection, and chronic inflammation [71,72], while immunosuppressive LDNs have been identified in septic shock [73], and multifaced LDNs were delineated in cancer. In addition, besides their possible role in these diseases and clinical conditions, the expression of specific cell-surface markers and their functional states still need to be coded.

## 3. Tumor-Associated Neutrophils (TANs)

### 3.1. Recruitment of Neutrophils in the TME

Neutrophil recruitment and survival in the TME involves the upstream regulation of myelopoiesis and a complex network of mediators, such as chemokines, cytokines, and complement components, which are produced and secreted by malignant, stromal, and immune cells present in the tumor niche [74,75]. Chemokines are the main drivers of this process and include the C-X-C motif chemokine ligands (CXCLs) CXCL1, CXCL2, CXCL5, CXCL6, and CXCL8 (also known as IL-8), which all promote neutrophil chemotaxis through the CXCR1 and CXCR2 receptors.

In lung cancer, tumor-derived CXCL1 contributes to TAN infiltration [76]. Similarly, in hepatocellular carcinoma, the upregulation of CXCL5 promotes tumor growth, lung metastasis, and intra-tumoral neutrophil infiltration, whereas its downregulation reduces tumor growth, metastasis, and TAN infiltration [77]. CXCL8 promotes tumor progression at multiple levels: by directly potentiating the migration and survival of cancer stem cells, by acting on endothelial cells to stimulate angiogenesis, and by inducing the trafficking of neutrophils and MDCSs which can thereby locally restrain anti-tumor immunity [78]. Beyond its effects on chemotaxis, CXCL8 can also have a major influence on neutrophil functions in ways that, at least theoretically, might have either positive or negative implications for tumor progression. CXCL8 induces the expression of Jagged2 (JAG2) on intra-tumoral neutrophils, and JAG2+ TANs suppress the cytotoxic activity of CD8+ T cells [79]. Interestingly, CXCL8 also induces granulocytic MDSCs to release NETs [80]. Although the significance of extracellular neutrophil-derived DNA in cancer patients is still unclear, previous reports have linked it to metastasis in the liver [81,82]. In addition to chemokines, inflammatory cytokines, including interleukin (IL)-17, IL-1β, and tumor necrosis factor (TNF), have all been implicated in neutrophil mobilization and recruitment to the cancer site. In particular, these cytokines are part of an inflammatory circuit that leads to the production of G-CSFs, and the subsequent formation and mobilization of neutrophils from the bone marrow. Moreover, IL-1β and G-CSF dramatically prolong the survival of neutrophils [83,84]. In addition, NSCLC with oncogenic KRAS expresses high levels of IL-17 and G-CSF, which attracts neutrophils, that in turn, mediate resistance to programmed death 1 (PD-1) therapy [85]. Along with G-CSF, granulocyte-macrophage colony stimulating factors (GM-CSFs) is another key molecule for the differentiation and mobilization of neutrophils from the bone marrow, and they are frequently secreted by tumors [84].

Neutrophils can also accumulate in the metastatic niche, where the expression of G-CSF, CXCL1, and CXCL2 by cancer cells and stromal cells promote their recruitment [2,86,87,88]. In an orthotopic transplantation model of breast cancer and a genetically engineered mouse model of oncogene-driven mammary carcinogenesis, the mobilization of neutrophils into the metastatic lung was regulated by the atypical chemokine receptor 2 (ACKR2), a decoy and scavenger receptor that is capable of binding the majority of inflammatory CC-chemokines expressed in early hematopoietic precursors [89]. Notably, Ackr2−/− mice showed enhanced tumor growth, but they were more protected against tumor metastasis through a neutrophil-dependent mechanism [89].

Altogether, these observations suggest that chemokines and other soluble factors exert direct effects on neutrophil recruitment and activation. However, the extent to which they contribute to the polarization of TANs in the TME is yet to have been elucidated.

### 3.2. Features of Tumor-Associated Neutrophils

During recent years, the heterogeneity of TANs in tumorigenesis has received careful attention [2,3,90]. However, the precise range of phenotypes and functions characterizing the TME, and how these phenotypes and functions impact tumor progression are still far from being delineated. Initially, based on the effects of the transforming growth factor-beta (TGF-β), murine TANs were classified as N1 (with anti-tumor activity) and N2 (with pro-tumor activity) [91]. The N1 phenotype was associated with the upregulation of TNF, CCL3, and intercellular adhesion molecule 1 (ICAM-1), along with the downregulation of the arginase axis, while the N2 phenotype was associated with high levels of various chemokines, including CCL2, CCL3, CCL4, CCL8, CCL12, CCL17, CXCL1, CXCL2, IL-8/CXCL8, and CXCL16 [92]. Although N1 and N2 tumor-associated neutrophils exhibit functional differences, definitive surface markers have not yet been identified [91]. Multiple subsequent analyses supported a pro-tumor role for TANs. However, the dynamics and the lifespan of neutrophils in the TME remain to be fully decoded, even if the ability of bone marrow neutrophils from mice with early-stage tumors to spontaneously migrate has been described. These cells lacked immunosuppressive activity, but had increased rates of oxidative phosphorylation and glycolysis, which led to a higher production of ATP compared to the control neutrophils [93]. In accordance with this observation, peripheral blood neutrophils from cancer patients exhibited an enhanced spontaneous migration and a greater response to CXCL8 or N-formyl-l-methionyl-l-leucyl-phenylalanine (fMLP) compared to the neutrophils obtained from healthy donors [93]. This indicates that neutrophils undergo dynamic changes during both cancer development and progression.

#### 3.2.1. Pro-Tumor Activity

TANs sustain tumor growth via different mechanisms that affect epithelial genetic instability, tumor cell proliferation, angiogenesis, tissue remodeling, and the suppression of innate and adaptive lymphoid cell-mediated immunity [11]. Neutrophils induce genetic instability via the production of ROS and through ROS-independent mechanisms, such as the release of microparticles containing specific pro-inflammatory microRNAs (miR-23A and miR-155), which promote the accumulation of DNA damage by downregulating the expression of molecules involved in the maintenance of nuclear integrity [94]. Then, TANs assist in tumor progression via the production of several cytokines and growth factors involved in tumor growth, including the epidermal growth factor, hepatocyte growth factor, and platelet-derived growth factors (PDGFs) [95,96].

Neutrophils are also involved in the early switch to angiogenic phenotype during tumorigenesis, as observed in both mice and humans [97]. In particular, TANs sustain tumor angiogenesis through the release of pro-angiogenic factors, such as prokineticin 2 (BV8), the chemoattractant S100 calcium-binding proteins S100A-8A and S100A9, and matrix metalloproteinase (MMP) 9. BV8 is mitogenic for endothelial cells, induces myeloid cell mobilization, and is involved in the resistance to anti-VEGF (vascular endothelial growth factor) cancer therapy [97,98,99]. S100A8/A9 are involved in cytoskeleton rearrangement and arachidonic acid metabolism as Ca^2+^ sensors [100]. Under physiological conditions, they are constitutively expressed in neutrophils, myeloid-derived dendritic cells, and monocytes. Under stress conditions, such as trauma, infection, and inflammation, S100A8/A9 are highly upregulated, and are secreted to modulate the inflammatory processes with the induction of leukocyte recruitment and cytokine secretion [101]. MMP9, also termed as gelatinase B, is the most complex member of the enzyme family of matrix metalloproteinases. It is released from neutrophils without its endogenous inhibitor and plays important roles in blood vessel growth. MMP9 degrades components of the extracellular matrix facilitating tissue remodeling, and activating growth factors, such as VEGF-A [102]. MMP9 is also contained in NETs, whose release takes place in many types of cancer.

NETs consist of chromatin DNA decorated with granule proteins released by neutrophils to trap microorganisms (Figure 2) [103,104,105]. A wealth of evidence supports the role of NETs in tumor growth and metastasis [106,107]. Indeed, cancer cells can be caught by NETs in the circulation and be stimulated to adhere to endothelial cells, invade the tissue, and reproduce at secondary sites [81,108,109]. Higher levels of NETs in the plasma and tumor tissues were observed in patients with advanced diffuse B-cell lymphoma and were found to be correlated with a dismal outcome in retrospective cohorts of patients [110]. NET infiltration was abundant in liver metastases of patients with breast cancer and in patients with colon cancer [79].

Moreover, serum NET levels are higher in patients with liver metastases compared to both patients without metastases and with metastases in other organs, thus indicating that levels of these NETs in the blood could act as a biomarker to precisely predict the long-term risk of liver metastases in patients with early-stage breast cancer [79]. The coiled-coil domain-containing protein 25 (CCDC25) plays a major role in this process [79]. Again, in patients with breast cancer, neutrophil infiltration and NETosis have been observed to increase more in lung metastases than in primary tumors. Moreover, higher NET levels were found in triple-negative versus luminal breast cancers [111]. NETs can also interact with platelets contributing to thrombosis, thereby also implying potential problems for organ dysfunction at non-metastatic sites in cancer patients [112]. Metastasis promotion is also improved through NET cooperation with ROS that has been shown to weaken epithelial barriers in favor of the metastatic spread of breast cancer cells in patients with a high body mass index [113].

As for MMP9, several other molecules promoting cancer cell proliferation are present inside the NET [114]. These include the high mobility group protein B1 (HMGB1), neutrophil elastase (NE), and MPO. HMGB1 is responsible for the activation of tumor cells via the toll-like receptor 9 (TLR9)-pathway, which enhances proliferation, migration, and the invasive potential of cancer cells. In the extracellular matrix, NE and MMP9 remodel laminin, whose cleavage activates the α3β1 integrin, which triggers the proliferation of cancer cells [82,115]. Circulating tumor cells (CTCs) can be trapped within the NET thereby stimulating metastasis formation. MPO in NETs stimulate endothelial cell proliferation, promoting a pro-angiogenic response via a hydrogen peroxide (H_2_O_2_)-induced TLR4 activation [116]. NE enhances cell proliferation through the activation of the phosphatidylinositol 3-kinase (PI3K) pathway triggered by the activation of the platelet-derived growth factor receptor (PDGFR) [117]. Furthermore, NE together with MPO positively regulate NET production with pro-tumoral effects.

The modulation of innate and adaptive immune cells represents another major mechanism through which neutrophils impinge tumor growth [118,119]. TANs can indeed suppress anti-tumor immunity using CD8+ T cells and natural killer (NK) cells. For example, IL-17 expression from γδ T cells results in the G-CSF-dependent accumulation of the TANs, along with the enhancement of their ability to suppress cytotoxic T lymphocytes [120]. Neutrophils mobilized by the primary tumors inhibit the NK cell–mediated clearance of tumor cells from the initial sites of dissemination and facilitate the extravasation of tumor cells into the lung parenchyma [120]. Ex vivo experiments demonstrated that neutrophils can suppress the tumoricidal activity of NK cells via a ROS-mediated mechanism [121].

The pivotal role of neutrophils in cancer progression is likely to be associated with the response to the ICIs. Neutrophils expressing ligands that activate the immune checkpoints present on the T cells and engage T cell exhaustion have been identified in different human and murine cancers. It has been reported that PD-L1 is expressed on neutrophils in human hepatocellular carcinoma and gastric carcinoma and correlates with a poor prognosis [122,123]. The V-domain immunoglobulin suppressor of T cell activation (VISTA) is highly expressed in neutrophils and MDSCs and is involved in the suppression of tumor-specific T cell responses and tumor evasion from the immune system [124]. Its blockade in a mouse model of melanoma was found to have induced an anti-tumor immune response [125]. However, high levels of immunosuppressive TANs in tumors may cause resistance to the VISTA blockade, thus suggesting that further investigations are needed to understand the impact of the VISTA on TAN activity in tumors. In lung cancer preclinical models, a high frequency of TANs was found to be correlated with a resistance to the PD-1 blockade, whereas the depletion of neutrophils with the IL-6-blocking antibody or a neutrophil-depleting Ly-6G-blocking antibody were determined to be able to reverse this phenomenon and enhance anti-tumor immune responses [85,126].

#### 3.2.2. Anti-Tumor Activity

Neutrophils can mediate different mechanisms of anti-tumor resistance, including direct cytotoxicity against tumor cells through ROS generation, and activation of T cell-dependent anti-tumor immunity or antimicrobial activity.

One of the best-known mechanisms of neutrophil-directed cytotoxicity against tumor cells is the production of ROS, such as superoxide ion and H_2_O_2_. Under conditions of hypoxia, neutrophils were found to kill cancer cells more effectively, especially through the production of ROS in a NADPH-oxidase dependent manner [127]. ROS-mediated toxicity may be dependent on the expression of an H_2_O_2_-dependent Ca^2+^ channel expression, namely TRPM2, which renders cancer cells to be more susceptible to neutrophil cytotoxicity [128]. Furthermore, neutrophils secrete reactive nitrogen species (RNS) like nitric oxide and peroxynitrite [34], whose direct effect on cancer cells is still unknown. However, in certain tumors with mutations or amplification of the MET proto-oncogene, MET itself is required for neutrophil chemoattraction and the release of nitric oxide, which in turn, promotes cancer cell killing [95]. While the production of ROS and RNS by neutrophils exhibits anti-tumor properties, a pro-tumor effect has linked to the ability of both ROS and RNS to elicit oxidative DNA damage, which in turn, causes genetic instability [129].

Beyond oxidative stress, NE and cathepsin G represent other molecules used by neutrophils to kill tumor cells. Indeed, neutrophils can secrete an active form of elastase which is able to proteolytically release the CD95 death domain-containing fragment, which, in turn, can kill malignant cells through a gain-of-function mechanism, indicating that elastase can attenuate primary tumor growth [130]. Furthermore, the killing capability of NE involves the suppression of survival pathways indicated by the decreased phosphorylation of JNK, ERK, and nuclear factor kB (NF-kB), the activation of the effectors of apoptosis with the enhancement of cleaved poly (ADP-ribose) polymerase (PARP) and caspase 3 (CASP3), and the induction of DNA damage as well as augmented mitochondrial ROS production [130]. In addition, NE can produce a CD8+ T cell-mediated abscopal effect to attack distant metastases [130]. Neutrophils express important factors related to apoptosis, such as the FAS ligand (FASL), TNF-related apoptosis-inducing ligand (TRAIL), and TNF. Among these, TRAIL, a type II membrane protein belonging to the TNF superfamily, is produced by different types of immune cells, including NK cells, monocytes, activated T cells, and neutrophils, but only in cancer cells is it able to induce apoptosis, since normal cells are resistant to TRAIL-mediated apoptosis [131]. The rapid release of TRAIL facilitates cancer cell clearance in particular tumor types and conditions like bladder cancer treated with BCG immunotherapy [132]. Aging neutrophils, after undergoing spontaneous apoptosis, also release the FASL [133,134]. Furthermore, activated neutrophils enhance the expressions of the TNF and FASL, resulting in their increased infiltration in a tumor area, and thus eventually leading to tumor cell clearance [135,136].

Neutrophils can control the metastatic progression by acting directly in the metastatic niche. The production of G-CSF and CCL2 by a primary tumor in a mouse model of breast cancer induced the recruitment and activation of neutrophils in the pre-metastatic lung along with the killing of cancer cells mediated by ROS. CCL2 also induced the recruitment of monocytes expressing CCR2 and producing interferon-γ (IFNγ) that enhanced the cytotoxic activity of neutrophils in the metastatic niche [137]. The anti-tumoral N1 phenotype has been associated with an upregulation of TNF, which promotes TAN priming for ROS release in order to enhance tumor cell death [138].

Along similar lines, neutrophils can kill cancer cells via trogocytosis and subsequent trogoptosis, which involves the disruption of cancer cell plasma membranes and their endocytosis by neutrophils [139,140]. Human neutrophils express FcαRI (also known as CD89) which exhibit a high affinity towards IgA, suggesting that they can kill IgA-opsonized cancer cells through ADCC [141,142].

Moreover, NETs can exert anti-tumoral activity as they are composed of molecules like MPO, toxic proteases, and histones, which can all inhibit tumor growth and metastasis [143]. In the blood of patients with head and neck squamous cell carcinoma (HNSCC), the presence of a specific neutrophil subset CD16^high^CD62L^dim^ indicated an improved cancer survival. This subset exhibited an increased NET activity, and the capacity to inhibit the migration, proliferation, and growth of HNSCC cells, thus suggesting that activated neutrophils exert direct anti-tumor activities that can be mediated by NETs [144].

In several early-stage cancers, TANs were shown to not be immunosuppressive but were shown to stimulate responses mediated by T cells through the expression of co-stimulatory molecules. In early-stage human lung cancers, a subset of TANs with characteristics of antigen–presenting cells (APCs), which expressed HLA-DR, CD14, CD206, CD86, and CCR7, were described. The frequency of APC-like TANs reduced as tumor size increased and became undetectable in large tumors [145]. This subset displayed the capacity to stimulate CD27^+^Ki67^high^PD-1^−^ T cells, trigger antigen-specific T-cell responses, and to uptake, degrade, and cross-present exogenous tumor antigens to effector CD8+ cells [145]. The accumulation of APC-like TANs in the T cell-rich zones of lymph nodes constitutes as a positive predictor for five-year survival [145]. TANs can stimulate the recruitment and activation of T cells in cancer through the production of several mediators, including the chemokines CXCL1, CXCL2, CXCL10, CCL2, and CCL3, respectively [146]. IFNγ and GM-CSF in the TME promote the maturation of immature neutrophils into APCs expressing HLA-DR and the co-stimulatory molecules CD86, 4-1BB ligand (4-1BBL), and OX40 ligand (OX40L), and are capable in amplifying the anti-tumor response mediated by the T cells [145]. Neutrophil deficiency has been associated with IFNγ production by a subset of unconventional CD4-CD8-αβ T cells in selected human tumors, including in undifferentiated pleomorphic sarcoma [147]. These unconventional subsets of T cells (like γδ T cells, mucosal-associated invariant T (MAIT) cells, and natural killer T (NKT) cells) also induce neutrophil differentiation into APCs for both the CD4^+^ and CD8+ T cells [148]. Furthermore, NKT cells indirectly prevent cancer progression by inhibiting the immunosuppressive neutrophils [149].

CXCL8, CXCL5, and CCL2, together with other stimuli, such as lipopolysaccharide (LPS) and IFNβ, or the inhibition of TGFβR signaling, enhance the oxidative burst in neutrophils and the consequent production of H_2_O_2_ which activates a signaling cascade in tumor cells, leading to TRPM2 activation and cell injury up to death due to a lethal influx of calcium (Ca^2+^) [91,128]. Another important mechanism of anti-tumor resistance exerted by neutrophils is their antimicrobial functions. In colorectal cancer, bacterial-driven inflammation and tumor development are reduced by the activation of the interleukin-1 receptor type 1 (IL-1R1) signaling pathway that takes place in neutrophils [150].

Other attempts of TANs to contrast tumor promotion were described in murine models of cancer. In an early-stages mouse model of uterine cancer, neutrophils were found to be able to oppose epithelial carcinogenesis through the induction of the basement membrane detachment of tumor cells and their death within the uterine lumen [151]. In a murine model of colitis-associated colon cancer, neutrophils were able to blunt colon tumor growth and invasion though the restriction of tumor-associated bacteria that were associated with a dramatic inflammatory response sustained by the bacteria-dependent IL-17 expression [152]. Finally, the innate immune training of granulopoiesis resulted in a potent anti-tumor activity. In mice bearing B16 melanoma, neutrophils were trained by the pre-treatment of mice with β-glucan, which induced the transcriptomic and epigenetic reprogramming of neutrophils towards an anti-tumor phenotype [153].

The above-mentioned observations indicate that neutrophils can exert a dual function in tumor immunity (Figure 3). The different type of cancer, the disease stage, and the complexity of tumor environment are key determinants of the acquisition of a pro-tumor or immunosuppressive phenotype of these cells, which determines their specific role in promoting or restraining cancer, respectively.

#### 3.2.3. Single-Cell Studies Resolving Neutrophil Heterogeneity

The introduction of single-cell approaches has revolutionized the field of basic research in immunology and cancer biology by providing high-resolution pictures of the TME in several malignancies, and also enabling the identification of multiple cell types/states for cell populations that were previously considered as homogeneous and mono-dimensional, including neutrophils [154]. Nonetheless, concerning neutrophils, various technical issues still need to be solved, as in most single-cell RNA sequencing (scRNA-seq) studies, these cells are clearly under-estimated, even though they should represent a significant proportion of infiltrating leukocytes, at least in certain tumors. However, recently, several landmark studies based on scRNA-seq data have been published [10,46].

In pancreatic ductal adenocarcinoma (PDAC), six PMN and four main TAN subclusters were identified [46]. Among these, TAN-1 can be considered as a pro-tumor factor, being characterized with a high expression of genes encoding for vascular endothelial growth factor A (VEGFA), urokinase plasminogen activator, which has a role in metastasis formation, and galectin 3 (LGALS3), which mediates the proliferation and stemness of cancer cells [155,156]. TAN-2 was considered as an inflammatory subset, with an elevated expression of the NLR family pyrin domain-containing 3 (NRLP3), phosphodiesterase-4B (PDE4B), and CD69 [46]. TAN-3 resembled PMN, thus indicating that it can represent a transitional stage of the neutrophil just homed in the TME. TAN-4 showed a unique transcriptional signature characterized by ISGs, including IFIT1, IFIT2, IFIT3, ISG15, and radical S-adenosyl methionine domain-containing 2 (RSAD2) [46].

In primary liver tumors, three clusters of neutrophils, two clusters of adjacent liver neutrophils, and six clusters of TANs were identified, respectively. Two TAN subsets expressed high levels of CCL3 and CCL4, and a subsequent analysis showed that CCL4+ TANs support macrophage recruitment. One TAN subset expressed elevated levels of CD274, encoding for programmed death ligand 1 (PD-L1), which can inhibit T cell cytotoxicity in vitro [4].

The Integration of several NSCLC single-cell datasets uncovered that tissue-resident neutrophils (TRNs) include three subsets of normal adjacent neutrophils (NANs) and four subsets of TANs [10]. Neutrophil maturity markers, including SELL (selectin L), prostaglandin-endoperoxide synthase 2 (PTSG2), CXCR2, CXCR1, Fc Gamma Receptor IIIb (FCGR3B), and membrane metalloendopeptidase (MME) were highly expressed in both populations. NAN and TANs signatures were clearly discernable through S100A12, which encodes a known marker of the activated pro-inflammatory neutrophils being specifically upregulated among the NANs [10]. Moreover, in TANs the gene encoding for peroxisome proliferator-activated receptor γ (PPARG), a transcription factor regulating oxidized LDL receptor 1 (ORL1), was found to have been upregulated. Interestingly, the TAN-2 subset was characterized by the expression of the major histocompatibility complex (MHC) class II genes HLA-DRA, CD74, HLA-DMB, and HLA-DRB1, thereby suggesting its role as antigen-presenting cells, whereas genes encoding for multiple cytokines, including CCL3, CCL4, cystatin B (CSTB), and LGALS3 were overexpressed in a TAN-3 subcluster [10].

Altogether, these observations underline the importance of high-resolution studies for TME classification, thus enabling an in-depth analysis of the main determinants to better stratify patients, and to find new specific targets which are useful for the development of new personalized therapies for those patients who are not responding to current treatments.

## 4. How the TME Can Influence Neutrophil Biology

The molecular mechanisms leading to neutrophil polarization remain largely unknown due to the uncertainty of whether mature neutrophils in circulation can be reprogrammed by environmental stimuli, like those present in the tissue and/or in the TME, or whether these defined phenotypes are programmed in the bone marrow. The evidence suggests that neutrophils are plastic cells, in that various neutrophil subsets exist in healthy tissues, and that these subsets adopt properties tailored to the needs of the specific tissue they home to [157]. This is especially relevant in the metabolically challenging environment of a tumor where several factors, including hypoxia, restricted nutrient availability, low pH, the presence of metabolites/oncometabolites and/or specific cytokines or chemokines, can modulate and/or interfere with the neutrophil’s phenotype, metabolism, and function [158,159].

During inflammation, neutrophils rapidly adapt to low oxygen levels in the tissues by stabilizing the hypoxia inducible factor (HIF)-1α and HIF-2α, which regulate neutrophil survival and other key functions [160,161]. Under hypoxic conditions, neutrophil apoptosis is inhibited and regulated by HIF-1α-dependent NF-kB activity [161]. Moreover, HIF-1α promotes the expression of PD-L1 on MDSCs, thus leading to the inhibition of T cell activation [162]. Since hypoxia is a common and chronic condition for tumor, stromal, and immune cells present in the tumor mass [159,160,161,162,163], it can therefore be suggested that a hypoxic TME could impact on the TANs phenotype, survival, and anti- or pro-tumor activity. However, conclusive evidence supporting whether and how hypoxia in the TME affects neutrophil biology are still lacking.

The TME forces malignant and immune cells, including neutrophils, to compete for fuels, and to face starvation or adaptation to metabolic stress [164]. Restricted nutrient availability also occurs either during physiological tissue growth, or because of different pathological processes, including wounds and/or vascular occlusions that lead to transient ischemia [165]. In the TME, starvation can occur during tumor growth due to poor vascularization, the massive use of nutrients by cancer cells, or as a result of chronic inflammation [165]. These conditions lead to the secretion of chemokines and cytokines, that, in turn, may promote the infiltration of immune cells, including neutrophils, suggesting that nutrient deprivation in the TME can serve as an initiator of inflammation within the tumor itself [165].

Moreover, the evidence suggests that neutrophils can adapt their metabolism in the presence of nutrient limitations. Indeed, in conditions where glucose and PPP-derived NADPH are inadequate, neutrophils engage in oxidative phosphorylation via c-Kit signaling to support the generation of ROS during the respiratory burst and to suppress T cell functions [50]. Other reports have shown that in human neutrophils, mitochondria can act as a direct source of ROS [166,167]. Moreover, neutrophils can oxidize lipids to compensate for the lack of glucose [54,168]. Consistently, circulating PMN-MDSCs from the head and neck, lung, or breast cancers accumulate more lipids than PMNs from healthy controls and can upregulate the fatty acid transport protein 2 (FATP2) involved in the uptake of arachidonic acid, and in the subsequent synthesis of prostaglandin E2 (PGE2), which, in turn, supports tumor growth and immune evasion [169]. Of note, the pharmacological inhibition of FATP2, alone or in combination with anti-cytotoxic T-lymphocyte antigen 4 (CTLA4), reduces tumor growth [169]. Circulating neutrophils from cancer patients and TANs can also uptake low-density lipoproteins (LDL) via lectin-type oxidized LDL receptor-1 (LOX-1) with direct implications in tumor progression [170]. LOX-1+ neutrophils have their gene signature, strong immunosuppressive activity, and other features characteristics of PMN-MDSCs [170]. Recent studies have indeed suggested that the TANs phenotype is characterized by the high expression of ORL1 (which encodes for LOX-1), VEGFA, CD83, ICAM1, and CXCR4, and the low expression of CXCR1, CXCR2, PTGS2, SELL, colony-stimulating factor 3 receptor (CSF3R), and Fc gamma receptor IIIb (FCGR3B) [10]. For this reason, LOX-1 has been described as a putative marker to distinguish the peripheral blood neutrophils from the PMN-MDSCs [170]. In addition to glucose and lipids, neutrophils also use high levels of glutamine under glucose-limiting conditions to support the functions of mitochondria and survival [47,171]. A role for glutamine in protecting neutrophils from apoptosis has also been reported in sepsis [172]. Along these lines, JHU083, a small molecule inhibiting glutamine catabolism along with OXPHOS induces apoptosis in both the tumor and in the circulation, thus improving the anti-tumor immunity in mice through different mechanisms, including the increase in the anti-tumor Th1 frequency, the upregulation of the IL2-STAT/mTORC1/Myc signaling pathways coupling with the glutamine downregulation, and the inhibition of the Th17 pro-tumor cells [173]. Furthermore, the glutamine blockade affects activated CD8+ T cells, whose metabolism shifts towards acetate metabolism to overcome tumor immune evasion [174].

Glycolytic malignant cells secrete lactate, which results in the acidification of the TME [175]. While T cells are more susceptible to a low pH, neutrophils appear to be more resistant, as in these conditions, neutrophil survival, chemotaxis, and endocytic capacity are enhanced rather than limited [176]. Whether lactic acid, produced by cancer cells as a by-product of aerobic or anaerobic glycolysis, might have a critical role in functional polarization of the TANs is still unknown.

## 5. How Neutrophils/TANs Can Influence Tumor Prognosis

High systemic levels of neutrophils, a high neutrophil-to-lymphocyte ratio (NLR), and an elevated infiltration of TANs in the tumor mass have all been associated with a poor prognosis in many types of cancer [90,119,177]. PMN-MDSCs have been discovered to accumulate in the blood of patients with advanced prostatic carcinoma, and the TAN signature has been associated with an unfavorable outcome, as neutrophil phenotypes are mostly consistent with a pro-tumorigenic role [178,179,180]. Consistently, pancreatic ductal adenocarcinoma (PDAC) has copious neutrophil infiltration [181,182], which has also been associated with a poor prognosis [183]. High NLRs in the peripheral blood has been deemed as a negative predictor of overall survival and disease-free survival [184], and levels of chemokines involved in neutrophil recruitment, including CXCL5, have been correlated with an advanced clinical stage and reduced survival [185].

Correlative studies in patients with NSCLC have associated NLRs with a worse prognosis and worse response to treatment [186]. The precise mechanism linking elevated NLRs and poor clinical outcomes in these patients is still undefined; however, evidence has shown that neutrophils are involved in the promotion of metastasis, whereas a decreased lymphocyte number is a biomarker of poor survival for patients with advanced cancer [187].

The NLR is also an important prognostic factor also advanced colon cancer [188]. A systematic review revealed that a pre-operative NLR > 5 is associated with a decreased long-term survival in patients with localized colorectal cancer (CRC), and in those with liver metastasis [189]. The NLR is a significant independent factor also influencing survival in patients with hepatocellular carcinoma (HCC) [190]. A growing body of evidence has indeed suggested that neutrophils participate in the pathogenesis of HCC at multiple levels: by inducing local immunosuppression, by direct enhancements in tumor cell survival, invasiveness, and metastatic capacity, and/or by remodeling the extracellular matrix to promote angiogenesis [191]. In primary liver cancers, although TANs are heterogenous, those expressing PD-L1 can suppress T cell cytotoxicity, and thus represent a promising target for immunotherapy [4].

Interestingly, accumulating evidence has sustained that neutrophil perturbations in cancer patients involve both neutrophils in the bone marrow and in the circulation. The committed unipotent early-stage NeP significantly increases tumor growth when transferred into murine cancer models, including in a humanized mouse model [28]. Moreover, human NeP was present in the blood of treatment-naive melanoma patients, but not in healthy subjects. The presence of a systemic cross-talk between the tumor and bones has been demonstrated, in which lung cancer can activate osteocalcin (Ocn)+ osteoblastic cells in the bone, and in turn, these cells supply tumor-infiltrating SiglecF^high^ neutrophils [192]. Similarly, a systemic inflammatory loop involving neutrophils has been described in advanced clear-cell renal-cell carcinoma [193], indicating that DNA demethylation drove cancer-cell intrinsic inflammation, with a subsequent increased transcription of the “inflammatory response”-related annotated genes, including genes encoding for the chemokine axis and serum amyloid A, and further affecting the neutrophil number and dynamics [193]. This means that cancer can act systemically to perturb the neutrophil quantity, heterogeneity, and functions.

In disagreement with the above results, in a number of human tumors, including CRC, low grade glioma, endometrial cancer, invasive ductal breast cancer, and undifferentiated pleomorphic sarcoma (UPS), high levels of TANs were discovered to be associated with a better prognosis [147,151,194]. In CRC, neutrophils enhance the responsiveness of CD8+ T cells to T cell receptor triggering [194]. In UPS, the neutrophil signature was associated with a type I immune response and an improved outcome [147]. In a mouse model of PTEN-deficient uterine cancer, neutrophils induced an impeded early-stage tumor growth, thus delaying malignant progression [151].

In summary, in the vast majority of tumors, neutrophils and MDSCs mediate vigorous systemic and tumor-localized immunosuppressive effects, thus impacting the tumor prognosis. However, TANs have also been associated with a better prognosis, thus indicating that their significance and functions can be heavily influenced by both the tissue and tumor microenvironments.

## 6. How Neutrophils/TANs Can Influence ICI Therapy

Immunotherapy with ICIs, mostly those targeting PD-1, PD-L1, or CTLA4, is now prevailing on chemotherapy for various malignancies. Available ICIs administered as monotherapies typically lead to objective response rates (ORRs) of around 20–40% in cohorts of patients with solid tumors, such as melanoma, NSCLC, or renal cell carcinoma, with a maximum response rate of 40–45% for melanoma and NSCLC, and an average response of about 20% for the others, respectively [193,195,196]. Several parameters linked to immune cells, tumor cells, and/or whole-body physiology can affect the clinical responses to the ICIs. In recent years, patient responses have been associated with the intra-tumoral infiltration of innate immune cells with immunosuppressive properties, including neutrophils, which are able to inhibit the recruitment and activation of T cells [197]. An abundance of preclinical and clinical data has demonstrated that a higher NLR is significantly associated with a decreased overall and progression-free survival, and also with lower rates of the responses and clinical benefits after ICI therapy in multiple malignancies [198].

A study performed on more than 500 patients with NSCLC, melanoma, renal cell carcinoma, head and neck cancer, bladder cancer, and sarcoma revealed that patients with a baseline NLR <  5 had a significantly longer OS after ICI [199]. On the contrary, NLR increase during treatment has also been associated with a poorer survival [199].

Prostate cancers are typically resistant to ICIs and express high levels of CXCL8 and other CXCR2 ligands [200,201]. In primary and metastatic castration-resistant prostate cancer (CRPC), ICIs alone was not sufficient in generating an efficient therapeutic response. Nonetheless, important synergistic responses were observed when ICIs were combined with MDSC-targeted therapy, which included cabozantinib and the phosphoinositide 3-kinase (PI3K)/mTOR dual inhibitor BEZ235 [202]. In pancreatic cancer, neutrophils and PMN-MDSCs mediate resistance to immunotherapy. The gain-of-function Trp53R172H mutation promotes the recruitment of neutrophils in the TME, which confers resistance to CD40 immunotherapy with gemcitabine and nab-paclitaxel chemotherapy [203]. Along similar lines, it has been found that IL-17-induced NETs generate resistance to PD-1/CTLA-4 therapy, thus suggesting that combining anti-IL17 with ICIs could represent a novel therapy with potential efficacy for PDAC [204].

In NSCLC, the ratio of CD8+ T cells and neutrophils within the tumor mass identified non-responsive patients to anti-PD1 monotherapy [205]. Preclinical evidence proved that neutrophils could restrict lymphocyte trafficking into the TME, thereby limiting the efficacy of PD-1 inhibitors and supporting the use of neutrophil-depleting drugs (like CXCR2 antagonists) in combination with ICIs [205]. Neutrophils express high levels of both CXCR1 and CXCR2. Inhibiting CXCR2 has been used to preclude neutrophil recruitment to tissues, and, currently, several clinical trials are investigating the safety and efficacy of small molecule inhibitors and antibodies targeting CXCR2 in combination with ICIs or radiotherapy in several types of cancer, including HNSCC, PDAC, metastatic castrate-resistant prostate cancer, hormone-sensitive prostate cancer, HCC, NSCLC, and metastatic melanoma [206]. Although CXCR2 is expressed on other cell types, including epithelial and endothelial cells, inhibitors of CXCR2 mostly impact neutrophils in tumor models [206].

The high-resolution single-cell atlas combining 22 datasets from almost 300 patients with NSCLC revealed that a gene signature derived from tissue-resident neutrophils (TRN) has a predictive and prognostic effect for patients treated with immunotherapy [10]. In particular, the TRN gene signature can identify patients refractory to treatment with atezolizumab [10]. The prognostic relevance for the anti-PD-L1 was stronger for lung squamous cell carcinoma compared to lung adenocarcinoma, respectively [10].

In conclusion, reducing the number of TANs and/or targeting specific interleukin/chemokines axis, including IL-8, in combination with immunotherapy or chemotherapy or targeted therapy may represent a novel treatment model to improve therapeutic responses across the different malignancies.

## 7. Conclusions

Neutrophils represent a large and heterogeneous population of cells with key roles in cancer growth, metastasis formation, and overall patients’ outcomes in multiple malignancies. However, the accumulating evidence supports an early anti-tumorigenic role for these cells. A more systematic effort integrating the state-of-the-art technologies, including single-cell and spatial transcriptomics, is needed to accurately determine the specific roles of neutrophils in different neoplasias. Deconvoluting the heterogeneity of TANs, in terms of their phenotype, metabolic aspects, and functions, and relating these aspects to patient prognosis and responses to therapy represent important, yet still unresolved, challenges. Although considerable advances have been made in understanding neutrophil biology, several outstanding aspects remain open questions. In particular: does heterogeneity also span metabolic aspects? Would targeting anti-tumorigenic neutrophil subsets to induce gain-of-function be a successful approach? If so, how can only anti-tumorigenic neutrophils could be targeted? Additional work is needed to answer these, and other questions, and to develop new strategies and therapies designed to complement the current options.

## Figures and Tables

**Figure 1 cancers-15-03327-f001:**
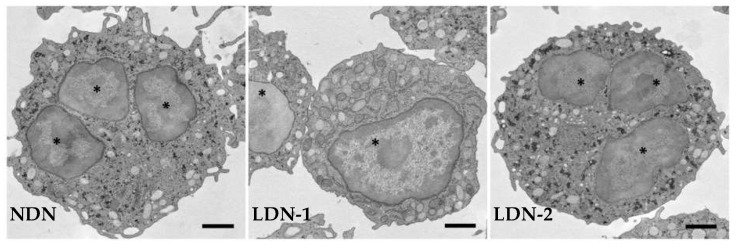
Representative transmission electron micrographs showing normal-density neutrophils (NDNs) and low-density neutrophils (LDNs) at different grades of maturation. LDNs are composed of immature cells with nuclei devoid of lobation, with a few granules, and intracellular vacuoles (LDN-1), and of mature cells with multi-lobed nuclei and a similar number of granules (LDN-2) as NDNs. * nucleus.

**Figure 2 cancers-15-03327-f002:**
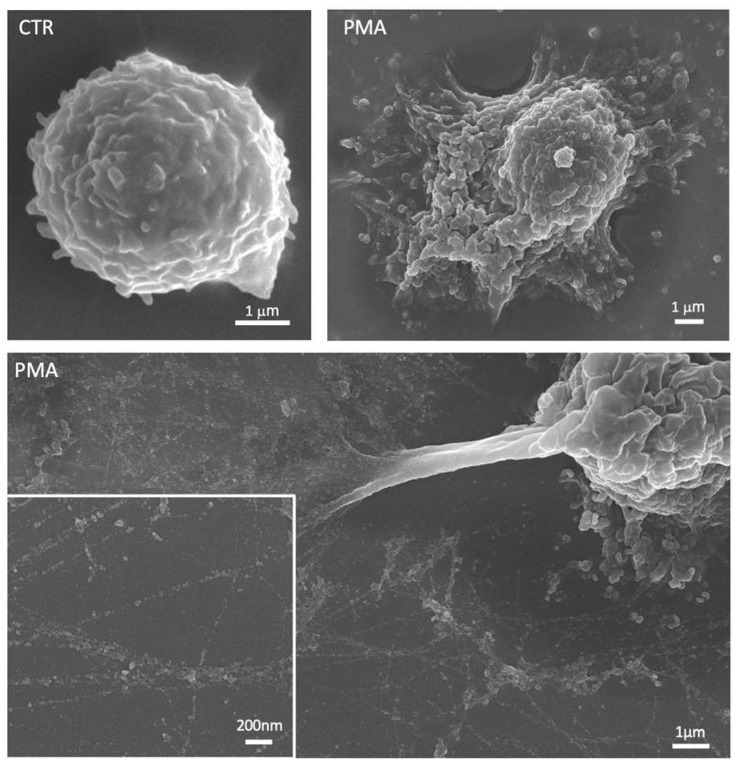
Representative scanning electron micrographs showing an unstimulated neutrophil (CTR) and neutrophils treated with phorbol 12-myristate 13-acetate (PMA). After stimulation, neutrophils are characterized by a flattened shape with cytoplasmic protrusions and numerous vesicles released in the extracellular space. The lower panel shows the neutrophil extracellular traps (NETs) with the smooth fibers and globular domains clearly visible in the insert at the higher magnification.

**Figure 3 cancers-15-03327-f003:**
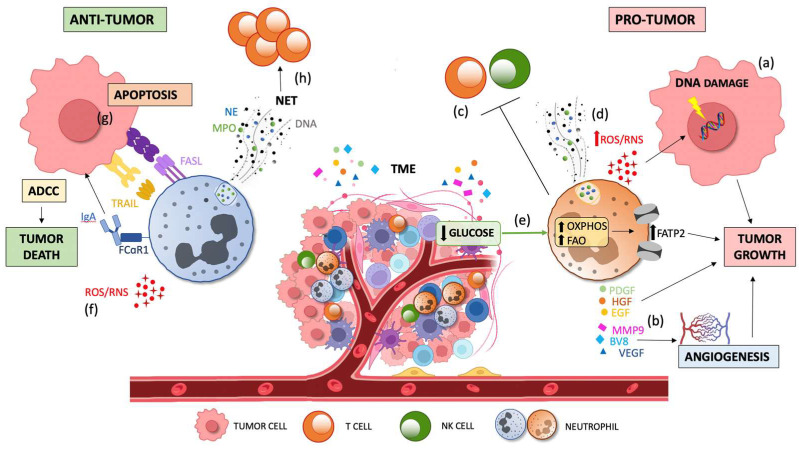
Dual contribution of TAN infiltration to tumor progression. Tumor growth is sustained by pro-tumor TANs via the production of ROS which induce DNA damage and genetic instability via the (**a**) production of cytokines and growth factors (EGF, HGF, and PDGF) and pro-angiogenic factors (BV8, MMP9, and VEGF), respectively; via the (**b**) suppression of innate and adaptive immunity; via the (**c**) promotion of metastasis through NET and ROS/RNS release and via (**d**) induction of oxidative phosphorylation and lipid accumulation by FATP2 upregulation due to the low glucose availability in the TME (**e**). Tumor death is promoted by anti-tumorigenic factors via the direct cytotoxicity against tumor cells through ROS and RSN generation; via (**f**) apoptosis induced by FAS-FASL and TRAIL signaling, and by the activation of ADCC through the binding of FcaRI to IgA; and via (**g**) lymphocyte activation by priming T cells mediated by NETs (**h**). TME, tumor microenvironment; TGFβ, transforming growth factor-beta; TANs, tumor-associated neutrophils; ROS, reactive oxygen species; RNS, reactive nitrogen species; EGF, epidermal growth factor; HGF, hepatocyte growth factor; PDGF, platelet-derived growth factor; MMP9, matrix metalloproteinase 9, VEGF, vascular endothelial growth factor; NET, neutrophil extracellular traps; FATP2, fatty acid transport protein 2; FASL, FAS ligand; TRAIL, TNF-related apoptosis-inducing ligand; ADCC, antibody-dependent cellular cytotoxicity; IgA, immunoglobulin A; and FcaRI, IgA Fc receptor.

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
