# Peer review of "Circulating and Tumor-Associated Neutrophils in the Era of Immune Checkpoint Inhibitors: Dynamics, Phenotypes, Metabolism, and Functions"

_cancers, 2023, doi:10.3390/cancers15133327_

Round 1
Reviewer 1 Report
This is a comprehensive and exhaustive review on the biology of neutrophils in cancer, covering various topics about these myeloid cells in several aspects of tumor progression. In particular, the authors focused on the impact of neutrophils in shaping the TME and in influencing tumor prognosis and response to ICI therapy. The review is very well written and clear. In some cases, the authors could include a more critical interpretation of the cited literature.
Specific comments:
- lines 71-81. Several steps of neutrophil development have been fully investigated and new intermediate populations have been identified in both mouse and human, such as preNeu and proNeu. The authors could briefly describe these subpopulations and their implication in cancer.
- lines 195-196 “However, their role in these diseases is still unresolved.” The authors dedicated a section of the review to LDNs. A conclusion focused on the roles of LDN and a critical interpretation of the literature could be added.
- lines 283-288. The role of S100A8 and S100A9 could be explained, as done for BV8 and MMP9.
- lines 322-324 “Nevertheless, NETs not only promote cancer metastasis, but have also proinflammatory antitumoral activity as they enhance lymphocyte activation by priming T cells and reducing their activation threshold [103].” This sentence could be moved to the paragraph 3.2.2 Anti-tumor activity. Given that NETs are primarily considered pro-tumoral, their anti-tumoral activity could be described more in details and critically.
- lines 337-345. In the section 3.2.1 Pro-tumor activity, the authors could extend the effect of neutrophils in inhibiting the adaptive immune response in cancer.
The suppression of the T cell response mediated by checkpoint receptor ligands expressed by neutrophils, such as PD-L1 and VISTA, could be included. Figure 3 could be implemented with this mechanism of tumor promotion.
- lines 364-369. The antitumor effects of NE could be described more in details.
- In the 3.2.2 Anti-tumor activity section, the interplay between neutrophils and unconventional T cells could be reported.
- lines 370-375. The role of neutrophil-associated FAS ligand, TRAIL and TNF could be better explained. Please, add references.
- line 535. Please specify that reference 144 refers to sepsis and not cancer.
- lines 535-538 “Along these lines, JHU083, a small molecule inhibiting glutamine catabolism and OXPHOS, induces apoptosis in both tumor and in the circulation, thus improving anti-tumor immunity in mice [145].” The authors should describe more in details this aspect.
- lines 575-580. Please, add references.
-lines 591-592 “In summary, neutrophils and MDSCs mediate vigorous systemic and tumor localized immunosuppressive effects, thus impacting tumor prognosis.” In this paragraph, the discussion is generally focused on NLR and TANs and their prognostic role. The sentence concluding the paragraph is not appropriate, since there are examples of tumors (e.g. colorectal cancer) in which TANs are associated with good prognosis. The entire section could be more equilibrated by presenting the complexity of using neutrophils as a prognostic factor.
- lines 637-641 “The high-resolution single-cell atlas combining 22 datasets from almost 300 patients with NSCLC revealed that a gene signature derived from tissue-resident neutrophils (TRN) has a predictive and prognostic effect for patients treated with immunotherapy [10]. In particular, the TRN gene signature identifies patients refractory to treatment with atezolizumab [10].” This study could be presented and discussed with more details.
- lines 642-645. As for the previous paragraph, the conclusions may be more critical, based on the complexity and heterogeneity of neutrophils in cancer.
Minor comments:
Some sentences should be revised, or simplified by dividing them.
e.g.:
- lines 36-37: One the possible mechanism
- lines 68-70: 1-2 x 1011
- 2.3 Metabolism: it should be “2.2 Metabolism”.
- lines 156-160. In this paragraph PMN-MDSCs are cited for the first time. The authors could add a sentence describing MDSC characteristics and their role in cancer, to help readers.
- Figure 1. The authors should add an indication of which images show NDN or LDN.
- lines 516-518 “Other reports showed that, in human neutrophils, mitochondria act as a direct source of ROS [138, 139], moreover neutrophils can oxidize lipids to compensate the lack of glucose [49, 140].” This sentence could be divided into two separate sentences.
Author Response
POINT-TO-POINT RESPONSE
Reviewer
This is a comprehensive and exhaustive review on the biology of neutrophils in cancer, covering various topics about these myeloid cells in several aspects of tumor progression. In particular, the authors focused on the impact of neutrophils in shaping the TME and in influencing tumor prognosis and response to ICI therapy. The review is very well written and clear. In some cases, the authors could include a more critical interpretation of the cited literature.
Author: We thank the reviewer for the comment.
Specific comments:
- lines 71-81. Several steps of neutrophil development have been fully investigated and new intermediate populations have been identified in both mouse and human, such as preNeu and proNeu. The authors could briefly describe these subpopulations and their implication in cancer.
Author: As suggested by the reviewer, we added the following description: “Neutrophil differentiation is a complex and heterogeneous process and occurs in the bone marrow (BM). It starts from hematopoietic stem cells (HSCs), that, differentiating into common myeloid progenitors (CMPs), give rise to the granulocyte-monocytes progenitors (GMPs) [15, 21]. The subsequent stages include promyelocyte, myelocyte, metamyelocyte, banded and segmented nucleus neutrophils. GMPs differentiate into promyelocytes, which express the neutrophil lineage marker CD66b [15, 21]. Subsequent neutrophil development occurs through the upregulation of CD11b and CD16. Promyelocytes differentiate into myelocytes and metamyelocytes, and lastly into banded (immature) and segmented (mature) neutrophils [14]. The traditional classification of neutrophil differ-entiation based on appearance may not represent the full process. Over the past decade, the use of single-cell technologies allowed a better definition of neutrophil maturation stages. In mice, GMPs develop into pro-neutrophils proNeu1, which develops into intermediate progenitors proNeu2, which in turn mature into highly proliferative pre-cursors pre-Neu. The last ones share a transcriptional profile with the unipotent neutrophil progenitors (NeP) described by Zhu et al., have little mobility and are most plentiful in the BM. They undergo further differentiation into non-proliferating immature neutrophils and subsequently mature neutrophils, which are predominantly present in blood, have lost their proliferative capacity, gained high motility and mediate of effector functions [11, 22]. Several studies have proposed that GMPs consist of several heterogeneous myeloid progenitor cells, including neutrophil progenitors, rather than a single homogeneous cell type. However, these studies do not exclude the possibility that an ‘earlier’, still unde-fined, progenitor exists that could give rise only to neutrophils and monocytes [7, 23-28]. The high degree of neutrophil heterogeneity is observed in cancer and is associated with disease progression. A typical hallmark of cancer is the egress of neutrophils from the bone marrow at earlier stages. It was observed that immature neutrophils with a banded nuclear morphology play a pro-tumorigenic role [11]. In lung cancer patients, devel-opmental stages even earlier than NeP/preNeu are present in blood and tumor [24]. In melanoma patients circulating progenitors NeP are increased compared to healthy sub-jects and sustain tumor growth and immunosuppression [28].”
- lines 195-196 “However, their role in these diseases is still unresolved.” The authors dedicated a section of the review to LDNs. A conclusion focused on the roles of LDN and a critical interpretation of the literature could be added.
Author: We added the following conclusion: “Although LDNs have been extensively studied in various diseases, several aspects of their functions, characteristics and phenotypes, are still a matter of intense controversy. They further enrich the concept of neutrophilic heterogeneity, as pro-inflammatory LDNs have been described in autoimmunity, infection and chronic inflammation [71, 72], immunosuppressive LDNs have been identified in septic shock [73], and multifaced LDNs were delineated in cancer. In addition, besides their possible role in those diseases and clinical conditions, the expression of specific cell-surface markers and their functional states still need to be coded.”
- lines 283-288. The role of S100A8 and S100A9 could be explained, as done for BV8 and MMP9.
Author: We added the following sentence: “S100A8/A9 are involved in cytoskeleton rearrangement and arachidonic acid metabolism as a Ca2+ sensor [100]. Under physiological conditions they are constitutively expressed in neutrophils, myeloid-derived dendritic cells, and monocytes. During stressing conditions, such as trauma, infection, and inflammation, S100A8/A9 are highly upregulated and secreted to modulate inflammatory processes with the induction of leukocyte recruitment and cytokine secretion [101].”
- lines 322-324 “Nevertheless, NETs not only promote cancer metastasis, but have also proinflammatory antitumoral activity as they enhance lymphocyte activation by priming T cells and reducing their activation threshold [103].” This sentence could be moved to the paragraph 3.2.2 Anti-tumor activity. Given that NETs are primarily considered pro-tumoral, their anti-tumoral activity could be described more in details and critically.
Author: The sentence has been moved to the paragraph 3.2.2. We also added the following paragraph “Moreover, NET can exert antitumoral activity as they are composed by molecules like MPO, toxic proteases, and histones that can inhibit tumor growth and metastasis [143]. In the blood of patients with head and neck squamous cell carcinoma (HNSCC), the presence of a specific neutrophil subset CD16highCD62Ldim indicated improved cancer survival. This subset exhibited an increased NET activity and the capacity to inhibit migration, pro-liferation, and growth of HNSCC cells, thus suggesting that activated neutrophils exert direct anti-tumor activity that can be mediated by NET [144].”
- lines 337-345. In the section 3.2.1 Pro-tumor activity, the authors could extend the effect of neutrophils in inhibiting the adaptive immune response in cancer.
The suppression of the T cell response mediated by checkpoint receptor ligands expressed by neutrophils, such as PD-L1 and VISTA, could be included. Figure 3 could be implemented with this mechanism of tumor promotion.
Author: As suggested by the reviewer we amended both text and figure 3. In particular, we added the following paragraph “The pivotal role of neutrophils in cancer progression is likely to be associated with the response to ICIs. Neutrophils expressing ligands that activate immune checkpoints on T cells and engage T cell exhaustion have been identified in different human and murine cancers. It was reported that PD-L1 is expressed on neutrophils in human hepatocellular carcinoma and gastric carcinoma and correlates with poor prognosis [122, 123]. V-domain immunoglobulin suppressor of T- cell activation (VISTA) is highly expressed in neu-trophils and MDSCs and is involved in the suppression of tumor-specific T cell responses and tumor evasion from the immune system [124]. Its blockade in a mouse model of melanoma induced an anti-tumor immune response [125]. However, high levels of immunosuppressive TANs in tumors may cause resistance to VISTA blockade, thus suggesting that further investigation is needed to understand the impact of VISTA on TANs activity in tumors. In lung cancer preclinical models, a high frequency of TANs correlated with resistance to PD-1 blockade, whereas the depletion of neutrophil with IL-6 blocking antibody or a neutrophil depleting Ly-6G blocking antibody, could reverse this phenomenon and enhanced anti-tumor immune response [85, 126].”
- lines 364-369. The antitumor effects of NE could be described more in details.
Author: We described the antitumor effects of NE, and we modified the paragraph as it follows “Beyond oxidative stress, NE and cathepsin G represent other molecules used by neu-trophils to kill tumor cells. Indeed, neutrophils can secrete an active form of elastase which is able to proteolytically release the CD95 death domain-containing fragment, which, in turn, can kill malignant cells through a gain-of-function mechanism, meaning that elastase can attenuate primary tumor growth [130]. Also, the killing capability of NE involves the suppression of survival pathways indicated by decreased phosphorylation of JNK, ERK and nuclear factor kB (NF-kB), activation of effectors of apoptosis with the enhancement of cleaved Poly (ADP-ribose) polymerase (PARP) and caspase 3 (CASP3), and induction of DNA damage as well as augmented mitochondrial ROS production [130].”
- In the 3.2.2 Anti-tumor activity section, the interplay between neutrophils and unconventional T cells could be reported.
Author: We reported the interplay between neutrophils and unconventional T cells. In particular, we added the following paragraph “Neutrophil deficiency has been associated with IFNγ production by a subset of uncon-ventional CD4-CD8- ab T cells, in selected human tumors including undifferentiated pleomorphic sarcoma [147]. These unconventional subsets of T cells (like gd T cells, mucosal-associated invariant T (MAIT) cells and natural killer T (NKT) cells) also induce neutrophil differentiation into APCs for both CD4+ and CD8+ T cells [148]. Furthermore, NKT cells indirectly prevent cancer progression by inhibiting immunosuppressive neutrophils [149].”
- lines 370-375. The role of neutrophil-associated FAS ligand, TRAIL and TNF could be better explained. Please, add references.
Author: We amended and we added the following paragraph “In addition, NE can produce a CD8+ T cell-mediated abscopal effect to attack distant metastases [130]. Neutrophils express important factors related to apoptosis, such as FAS ligand (FASL), TNF-related apoptosis inducing ligand (TRAIL) and TNF. Among these, TRAIL, a type II membrane protein belonging to TNF superfamily, is produced by different types of immune cells including NK cells, monocytes, activated T cells, and neutrophils, but only in cancer cells it is able to induce apoptosis, since normal cells are resistant to TRAIL-mediated apoptosis [131]. The rapid release of TRAIL facilitates cancer cells clearance in particular tumor types and conditions like bladder cancer treated with BCG immunotherapy [132]. Aging neutrophils, after undergoing spontaneous apoptosis, also release FASL [133, 134]. Also, activated neutrophils enhance the expression of TNF and FASL resulting in their increased infiltration in a tumor area, thus eventually leading to tumor cell clearance [135, 136].”
- line 535. Please specify that reference 144 refers to sepsis and not cancer.
Author: we specified that ref 144 (now ref 172) refers to sepsis.
- lines 535-538 “Along these lines, JHU083, a small molecule inhibiting glutamine catabolism and OXPHOS, induces apoptosis in both tumor and in the circulation, thus improving anti-tumor immunity in mice [145].” The authors should describe more in details this aspect.
Author: We described more in details this aspect. We added the following sentence: “Along these lines, JHU083, a small molecule inhibiting glutamine catabolism and OXPHOS, induces apoptosis in both tumor and in the circulation, thus improving an-ti-tumor immunity in mice through different mechanisms, including the increase of an-titumor Th1 frequency, the upregulation of IL2-STAT/mTORC1/Myc signaling pathways coupling with the glutamine downregulation, and the inhibition of Th17 pro-tumor cells [173]. Also, glutamine blockade affects activated CD8+ T cells whose metabolism shifts towards acetate metabolism to overcome tumor immune evasion [174].”
- lines 575-580. Please, add references.
Author: We added the reference. It is ref 28.
-lines 591-592 “In summary, neutrophils and MDSCs mediate vigorous systemic and tumor localized immunosuppressive effects, thus impacting tumor prognosis.” In this paragraph, the discussion is generally focused on NLR and TANs and their prognostic role. The sentence concluding the paragraph is not appropriate, since there are examples of tumors (e.g. colorectal cancer) in which TANs are associated with good prognosis. The entire section could be more equilibrated by presenting the complexity of using neutrophils as a prognostic factor.
Author: We modified accordingly. In particular we added the following paragraph “In disagreement to the above results, in a number of human tumors, including CRC, low grade glioma, endometrial cancer, invasive ductal breast cancer, and undifferentiated pleomorphic sarcoma (UPS), high levels of TANs were associated with better prognosis [147, 151, 194]. In CRC, neutrophils enhance the responsiveness of CD8+ T cells to T-cell receptor triggering [194]. In UPS, neutrophil signature was associates s were associated with a type I immune response and an improved outcome [147]. In a mouse model of PTEN-deficient uterine cancer, neutrophils induced impeded early-stage tumor growth, thus delaying malignant progression [151].
In summary, in the vast majority of tumors, neutrophils and MDSCs mediate vig-orous systemic and tumor-localized immunosuppressive effects, thus impacting tumor prognosis. However, TANs have also been associated with better prognosis, thus indi-cating that their significance and functions can be heavily influenced by both tissue and tumor microenvironments.”
- lines 637-641 “The high-resolution single-cell atlas combining 22 datasets from almost 300 patients with NSCLC revealed that a gene signature derived from tissue-resident neutrophils (TRN) has a predictive and prognostic effect for patients treated with immunotherapy [10]. In particular, the TRN gene signature identifies patients refractory to treatment with atezolizumab [10].” This study could be presented and discussed with more details.
Author: We presented and discussed the study by Salcher et al in other parts of the review. In section 3.2.3, more than a paragraph was dedicated to the study. The sentence reported by the reviewer was in section 6 and was about TRN/TAN signatures in patients treated with ICIs. However, we added the following sentence “The prognostic relevance for the anti-PD-L1 was stronger for lung squamous cell carcinoma compared to lung adenocarcinoma [10].”
- lines 642-645. As for the previous paragraph, the conclusions may be more critical, based on the complexity and heterogeneity of neutrophils in cancer.
Author: As requested by the reviewer, we modified the conclusion as it follows “Neutrophils represent a large, heterogeneous, population of cells with key roles in cancer growth, metastasis formation, and overall patients’ outcomes in multiple malignancies. However, accumulating evidence supports an early anti-tumorigenic role for these cells. A more systematic effort integrating state-of-the-art technologies, including single-cell and spatial transcriptomics, is needed to accurately determine the specific roles of neutrophils in different neoplasias. Deconvoluting the heterogeneity of TANs, in terms of phenotype, metabolic aspects and functions, and relating these aspects to patient prog-nosis and response to therapy represent important, still unresolved, challenges. Although considerable advances have been made in understanding neutrophil biology, several outstanding aspects remain open questions. In particular: does heterogeneity spans also metabolic aspects? Would targeting anti-tumorigenic neutrophil subsets to induce gain-of-function be a successful approach? If so, how only anti-tumorigenic neutrophils could be targeted? Additional work is needed to answer these, and other, questions, and to develop new strategies and therapies designed to complement current options.”
Minor comments:
Some sentences should be revised, or simplified by dividing them.
e.g.:
- lines 36-37: One the possible mechanism
Author: We modified.
- lines 68-70: 1-2 x 1011
Author: We modified.
- 2.3 Metabolism: it should be “2.2 Metabolism”.
Author: We modified accordingly.
- lines 156-160. In this paragraph PMN-MDSCs are cited for the first time. The authors could add a sentence describing MDSC characteristics and their role in cancer, to help readers.
Author: We modified as requested.
- Figure 1. The authors should add an indication of which images show NDN or LDN.
Author: We added NDN or LDN on images.
- lines 516-518 “Other reports showed that, in human neutrophils, mitochondria act as a direct source of ROS [138, 139], moreover neutrophils can oxidize lipids to compensate the lack of glucose [49, 140].” This sentence could be divided into two separate sentences.
Author: We modified.

Reviewer 2 Report
It is an exciting review focusing on the heterogeneity of neutrophils and immune checkpoint inhibitors. The review presents an interesting idea concerning neutrophils relevance on cancer. However, some points can be better discussed.
Major revision
The authors should check for missing references. There are paragraphs without a reference.
The authors should specify in the text when the referenced studies are about human or murine neutrophils.
The authors mention that there are few papers in the literature focusing on human neutrophils. One way to elaborate more on the subject is to look for papers that show the polarization of human neutrophils in vitro.
The figures presented must have their references.
Minor Revision
Some symbols have changed in the text. They must be checked.
Author Response
POINT-TO-POINT RESPONSE
It is an exciting review focusing on the heterogeneity of neutrophils and immune checkpoint inhibitors. The review presents an interesting idea concerning neutrophils relevance on cancer. However, some points can be better discussed.
Major revision
The authors should check for missing references. There are paragraphs without a reference.
Author: We added missing references.
The authors should specify in the text when the referenced studies are about human or murine neutrophils.
Author: As suggested by the reviewer, we specified when studies are about human or murine neutrophils.
The authors mention that there are few papers in the literature focusing on human neutrophils. One way to elaborate more on the subject is to look for papers that show the polarization of human neutrophils in vitro.
Author: We thank the reviewer for the comment. We reported papers regarding polarization of human neutrophils in vitro. However, our sentence was referred to the fact that there are few studies on neutrophils heterogeneity and on the mechanisms by which this heterogeneity is established and maintained.
The figures presented must have their references.
Author: There are not references for figures as they were produced in our lab and are unpublished. Otherwise, we can remove figure 1 and figure 2.
Minor Revision
Some symbols have changed in the text. They must be checked.
Author: Symbols have been checked.
